# On-Road Evaluation of an Unobtrusive In-Vehicle Pressure-Based Driver Respiration Monitoring System

**DOI:** 10.3390/s25092739

**Published:** 2025-04-26

**Authors:** Sparsh Jain, Miguel A. Perez

**Affiliations:** 1Division of Data and Analytics, Virginia Tech Transportation Institute, 3500 Transportation Research Plaza, Blacksburg, VA 24061, USA; mperez@vtti.vt.edu; 2Department of Biomedical Engineering and Mechanics, Virginia Tech, Blacksburg, VA 24061, USA

**Keywords:** respiration monitoring, driver monitoring, in-vehicle sensing, automotive safety, occupant detection, post-crash triage, continuous health monitoring, vital signs

## Abstract

**Highlights:**

**What are the main findings?**
Non-intrusive pressure-based sensors integrated into vehicle seats can reliably estimate the respiration rate in a moving vehicle, even amidst significant noise and artifacts.Experimental evaluation during on-road driving demonstrates accurate performance for real-world applications, effectively closing the gap with practical deployment.

**What is the implication of the main finding?**
These results enable the development of unobtrusive driver and occupant monitoring systems for real-time, in-vehicle health and safety assessments, which are critical for post-crash care and impairment detection.

**Abstract:**

In-vehicle physiological sensing is emerging as a vital approach to enhancing driver monitoring and overall automotive safety. This pilot study explores the feasibility of a pressure-based system, repurposing commonplace occupant classification electronics to capture respiration signals during real-world driving. Data were collected from a driver-seat-embedded, fluid-filled pressure bladder sensor during normal on-road driving. The sensor output was processed using simple filtering techniques to isolate low-amplitude respiratory signals from substantial background noise and motion artifacts. The experimental results indicate that the system reliably detects the respiration rate despite the dynamic environment, achieving a mean absolute error of 1.5 breaths per minute with a standard deviation of 1.87 breaths per minute (9.2% of the mean true respiration rate), thereby bridging the gap between controlled laboratory tests and real-world automotive deployment. These findings support the potential integration of unobtrusive physiological monitoring into driver state monitoring systems, which can aid in the early detection of fatigue and impairment, enhance post-crash triage through timely vital sign transmission, and extend to monitoring other vehicle occupants. This study contributes to the development of robust and cost-effective in-cabin sensor systems that have the potential to improve road safety and health monitoring in automotive settings.

## 1. Introduction

Unobtrusive physiological sensing in automobiles has gained considerable momentum due to its potential to improve road user safety, enhance driver well-being, and protect vulnerable passengers such as children and infants [1,2,3,4]. This technology, however, must demonstrate robust performance and reliability in on-road scenarios before it can be applied to a range of applications, such as post-crash occupant triage (communicating real-time vitals to emergency responders [5,6,7,8]), driver monitoring systems for fatigue, distraction, drowsiness, and impairment detection [1,9,10,11,12], child presence alerts to prevent heatstroke, and applications in commercial vehicles, where monitoring driver health and alertness is key to preventing crashes and ensuring safe operation during long hours on the road [13]. Additionally, as autonomous driving technology advances, physiological monitoring systems will be essential for assessing driver readiness to regain control when needed [2,14].

A variety of approaches for respiration monitoring have been proposed in the existing literature. For example, camera-based methods rely on standard or infrared video to detect small facial or chest movements and temperature changes [15,16,17,18,19,20,21]. However, although camera-based approaches may be effective under controlled conditions, their performance may degrade if their line of sight is blocked or if the lighting varies widely [22]. In contrast, radar-based systems leverage the Doppler effect caused by chest motion. These systems, however, can be sensitive to occupant position and ambient interference [23,24,25,26]. Ultrasound sensors, in turn, similarly detect chest displacement and have shown promise in clinical and home-care contexts; however, bulky transducers and stringent alignment requirements often limit their practicality in dynamic automotive settings [27,28,29,30].

In contrast, pressure-based sensing, which is already integrated into many vehicle seats to measure vehicle occupancy, may offer a suitable alternative [31,32]. Many vehicle occupant classification modules (OCMs) use pressure sensing systems that incorporate fluid-filled bladders or embedded pressure mats in car seats. These pressure sensors are used to detect the presence of passengers and classify occupant size for proper airbag deployment [33]. However, in recent years, researchers have adapted these seat-integrated modules to capture subtle pressure changes produced by the cardiopulmonary cycle [18,34]. In the automotive context, Wusk and Gabler [34] initially demonstrated that existing fluid-filled seat bladders could be repurposed to detect respiration in resting subjects in a controlled laboratory environment. A subsequent study conducted by Valente et al. [18] developed an enhanced system that worked in a stationary vehicle with the engine running. This study demonstrated the feasibility of measuring a passenger’s breathing rate despite moderate noise and vibration, paving the way for a more advanced design suitable for on-road scenarios while highlighting the need for robust filtering and artifact rejection to mitigate noise.

Building on these foundational studies, the present work addresses the key gap of implementing a pressure-based respiration-sensing approach in a real-world moving vehicle, specifically verifying system performance under real driving conditions, where noise and driver movement are substantially higher than in a laboratory setting. The work focuses on the vehicle’s driver, since this occupant represents a more complex sensing challenge than passengers due to required movements related to steering and braking, which can significantly overshadow the subtle waveform of respiration [35]. These dynamic movements, along with the vehicle’s motion, create a challenging environment for accurate physiological sensing.

The subsequent sections of this paper detail the approach for capturing respiration signals in a moving vehicle, outline the signal processing strategies, present the results, and discuss how these findings could generalize to other occupant monitoring scenarios.

## 2. Materials and Methods

### 2.1. Experimental Setup

This test-track driving study was part of a pilot investigation that involved five participants (3 males, 2 females). Each participant wore a heart rate monitor (Polar H10 by Polar Electro, Kempele, Finland),an electroencephalograph (EEG) headset (DSI-24 EEG by Wearable Sensing, San Diego, CA, USA), and a respiration monitoring belt (Wearable Sensing DSI-RESP). These sensors recorded continuous data during the experimental sessions, which included a baseline driving session (S1), an alcohol dosing session (D1), and an impaired driving session (S2). The two driving sessions (S1 and S2) were identical. Participants drove a 2018 Ford Edge SE (Ford Motor Company, Dearborn, MI, USA) retrofitted with rear-seat driving controls for safety intervention on the Virginia Smart Roads, a closed-to-the-public test track. Each driving session lasted 20 to 40 min and involved a series of maneuvers such as left and right turns, lane changes, and stops at traffic signals. The route included an urban intersection section (25 m.p.h. speed limit), followed by a road segment (35 m.p.h. speed limit). Participants received continuous navigation instructions from an experimenter seated behind them and were instructed to follow all traffic laws and use turn signals when appropriate.

The front passenger seat of the research vehicle was retrofitted with an original equipment manufacturer (OEM) fluid-filled pressure-sensing bladder and sensor assembly from a 2012 Ford Taurus SE (Figure 1a–c), as this is a part of many OCM systems used in consumer vehicles (see Table 1 for details). Previous studies established that this assembly is able to detect the respiration rate through minute pressure changes caused by inhalation and exhalation cycles [18,34]. The sensor output (0–5 V DC corresponding to the detected pressure) was recorded via a USB-connected LabJack U3 HV analog-to-digital converter (ADC) at a sampling rate of 50 Hz. The DSI RESP belt, which served as the reference device, was connected to the DSI EEG headset and transmitted data through Bluetooth Low Energy at 300 Hz. Data from both devices were time-synchronized and recorded into Extensible Data Format (XDF) files using Lab Streaming Layer (LSL version 1.16) [36]. LSL was chosen for this study due to its ability to facilitate multimodal data collection from sensors operating at different and potentially irregular sampling rates while maintaining sub-millisecond synchronization. As an open-source middleware ecosystem, LSL simplifies data streaming, synchronization, and recording by providing a standardized application programming interface (API) that abstracts platform differences and ensures precise time alignment across diverse physiological signals. Custom LSL packages were developed by modifying open-source device-specific streaming packages (’dsi2lsl’ [37] for the EEG/RESP sensor and ’labjack-to-lsl’ [38] for the Labjack U3 ADC) to stream the data.

Seat sensor recording began just before the participant started driving and ended once the session was completed. Thus, for each participant, two datasets (S1 and S2) were collected, for a total of 10 datasets among the five participants. During the sessions, participants engaged in normal driving activities, including occasional verbal communication, without restrictions on body movements. This approach was intended to replicate the noise and movement-related artifacts expected in real-world driving conditions.

### 2.2. Signal Processing

The normal respiration rate for healthy, resting adults ranges from 12 to 18 breaths per minute (brpm). This study aimed to capture respiration rates in the broader range of 6–30 brpm, including potentially abnormal values. The sensor output was sampled at a frequency of 50 Hz to comfortably exceed the Nyquist rate for the target frequency band (0.1–0.5 Hz), ensuring sufficient resolution for capturing respiration dynamics. This high sampling rate also enabled detection of potential high-frequency disturbances and alignment with other synchronized systems in the vehicle, though these secondary factors were not relevant to the current analysis. All data processing and analyses were designed with this target range in mind and were implemented post hoc in MATLAB R2023a [39]. The raw signals from the respiration belt and pressure sensor were filtered using a zero-phase Infinite Impulse Response (IIR) bandpass filter in the 0.1–0.5 Hz range (filter specifications are provided in Table 2) [40]. This filter was designed to isolate respiration rates within the 6 to 30 brpm range, and its frequency response is shown in Figure 2. The filtered signals were then normalized to an amplitude range of 0 to 1 for improved visualization.

Signal peaks were detected using the following criteria:

In a discrete set of *n* samples $\left(s_{1},s_{2},\cdots ,s_{n}\right)$, the *j*th sample (j∈N,2≤j≤n−1) is initially classified as a signal peak if it satisfies two conditions:$s_{j}>s_{j-1}$ (the sample is greater than the previous value, indicating an upward slope) and$s_{j}\geq s_{j+1}$ (i.e., the sample is greater than or equal to the next value, ensuring a local maximum).

Figure 3 demonstrates the condition being satisfied for the *j*th sample in a theoretical dataset. After identifying all initial peaks, the algorithm applies a minimum peak distance threshold of 2 s, corresponding to the physiological limit of 30 brpm. This threshold allows for dynamic evaluation of peaks within the window, selecting only the most prominent peak while ignoring others within the same time frame. This approach effectively retains only the strongest peak, minimizing the influence of small fluctuations and reducing false positive peaks. Figure 4 illustrates the steps and final output of this process for a 60-s period involving a single participant. While the raw seat pressure signal exhibits voltage variations on the order of 10^−1^ V, the filtered signal appears an order of magnitude smaller, typically within the 10^−3^ V range (see Figure 4b). This low-amplitude result is expected and reflects the nature of respiration-induced pressure changes, which are subtle and heavily damped as they propagate through the body, clothing, and seat materials. It is important to note that the original sensor was not designed for respiration monitoring; as such, these oscillations are often treated as noise in occupant classification contexts. Here, however, the same fluctuations constitute the signal of interest. As such, traditional signal-to-noise ratio (SNR) interpretations are not directly applicable; the signal’s amplitude is inherently small relative to the broadband noise it is embedded in, but this does not indicate poor sensing fidelity. The processing pipeline effectively isolates the relevant low-frequency component, allowing robust respiration rate estimation even from the low-amplitude input.

Finally, respiration rates for each sensor are calculated by precisely counting the number of peaks within a 60-s sliding window, with a 30-s step size (i.e., 50% overlap) across the entire trial period. Fractional cycles are also accounted for by considering both the peaks preceding and succeeding each epoch, ensuring an accurate estimation of the respiration rate.

### 2.3. Data Analysis

Mean respiration rates and standard deviations were calculated for each sensor and trial. These values were analyzed as dependent variables using a random-effects mixed general linear model in JMP Pro 18. The session (S1 or S2) and sensor types (belt or pressure sensor) were treated as independent variables. Significance was assessed for the main effects and two-way interactions with an alpha level of 0.05. Post hoc analyses were conducted using Tukey’s Honestly Significant Difference (HSD) test for multiple comparisons.

Bland–Altman analysis was additionally performed to assess the performance of the pressure sensor compared to the true respiration rates. This statistical method is used to assess the agreement between two measurement techniques and is particularly useful for comparing a new measurement technique against an established reference, as it highlights potential discrepancies and systematic biases that may not be apparent through correlation analysis alone [41]. It plots the mean of the two methods against their difference, allowing for visualization of systematic bias and limits of agreement (LoAs). The LoAs, which are typically set at a mean difference of ±1.96 standard deviations, represent the expected range within which most differences should lie.

Rate differences (i.e., d=Truerate−Measuredrate) were calculated to derive the following metrics:Mean of differences,Mean absolute error (MAE),Standard deviation of differences (SD), andCoefficient of variation $\left(CV=\frac{\mathrm{SD}}{\mathrm{True}\;\mathrm{mean}\;\mathrm{rate}}\times 100\%\right)$.

Boxplots and statistical summaries were generated for individual subjects, as well as the entire dataset.

#### Additional Analysis

All primary results were obtained using a 60-s sliding window with a 30-s step size (50% overlap). This configuration was selected as a practical compromise between estimation stability and real-time responsiveness, particularly for steady-state driver monitoring. To assess system adaptability, additional analyses were conducted using a range of window lengths and step sizes. Table 3 summarizes the parameter combinations explored. These configurations were evaluated not to identify a single optimal setup but to examine how performance varies under different operational demands, such as rapid estimation in emergency contexts versus smoother trend tracking for continuous monitoring.

## 3. Results

Five healthy adult participants (3 male, 2 female) between the ages 27 and 46 years (mean = 34.8 years) successfully completed the study. Table 4 summarizes their demographics.

The processed data were analyzed to determine the accuracy of respiration rate estimation using the pressure sensor. The following subsections detail the time-domain results, statistical analysis, and additional evaluations.

### 3.1. Time-Domain Results

Figure 4 in Section 2.2 illustrates the comparison between processed respiration signals from both sensors over a randomly selected 60-s period. The normalized waveforms are overlaid to facilitate a visual assessment of the detected peaks. To provide a more detailed view, Figure 5 zooms in on a 30-s segment, offering clearer insight into how accurately the pressure sensor tracks the true respiration rate.

The respiration rate values derived from the pressure sensor were compared to the true respiration rate (Figure 6a illustrates the result for a randomly selected 25-min period). In the period illustrated in Figure 6a, the pressure sensor maintains a fairly accurate estimation throughout this duration. A slight increase in true respiration rate between 0 and 5 min and a slight decrease around the 18-minute mark were successfully captured by the pressure sensor. Figure 6b presents the fluctuation of the measurement error around the true respiration rate, consistently staying within ±2 brpm in most cases, with occasional deviations reaching ±3 brpm in noisier segments.

### 3.2. Statistical Results

The respiration rate values from the pressure sensor and the reference device were analyzed for all five participants. The measured respiration rates showed general agreement with the true respiration rates, although individual variations were observed (Figure 7). The summary statistics for each subject, including the duration and percentage of the data used in the analysis, mean error, MAE, and SD, are presented in Table 5.

Overall performance metrics indicated that, on average, the pressure sensor estimated respiration rates with an MAE of 1.5 brpm. The mean measured respiration rate was only 0.05 brpm higher than the true respiration rate. The standard deviation of the measurement error was found to be 1.87 brpm, corresponding to a 9.2% CV relative to the true mean respiration rate over the entire dataset.

The generalized linear model (as described in Section 2.3) yielded least square mean values of 19.0 and 19.3 brpm for the true and measured respiration rates, respectively. Standard errors for the belt and seat sensor were both found to be nearly identical at approximately 0.51 brpm, while their individual standard deviations were 1.58 brpm and 1.21 brpm, respectively. The difference in the mean breathing rates detected by the two sensors was not statistically significant (*F* = 0.411, *p* = 0.557).

#### Bland–Altman Analysis Results

The measured respiration rate values were in good agreement with the true values, with approximately 67% of all measured values falling within ±1 standard deviation of the rate difference (Figure 8). The reproducibility coefficient (RPC), determined by 1.96×SD, was 3.7 brpm, which mostly fell within acceptable limits.

### 3.3. Additional Results

Different window configurations exhibited different predictive performance, with longer time windows generally resulting in lower errors (Figure 9 and Table 6). Worst-case performance was observed with a 20-s window with 50% overlap, where the MAE reached 2.4 brpm. This level of accuracy may not be ideal for applications requiring high precision but is acceptable for cases where a rapid respiration rate estimation is needed, such as post-crash response scenarios. A 60-s window length with 50% overlap produced an MAE of 1.5 brpm. Further improvements were seen with 90-s and 120-s windows, achieving near-perfect performance with errors close to 1 brpm.

## 4. Discussion

This study demonstrates that the proposed system can reliably estimate the respiration rate in real-world driving conditions. With an MAE of 1.5 brpm and a standard deviation of 1.9 brpm, the system offers sufficient accuracy for applications such as driver monitoring, post-crash triage, and passenger safety. The Bland–Altman analysis further supports the validity of this approach, showing strong agreement with the true respiration rates and minimal measurement bias. Furthermore, the ability of this approach to function in a moving vehicle while maintaining accuracy reinforces its viability for unobtrusive, real-time monitoring. The sensor’s integration within the seat enables continuous monitoring as long as the occupant remains seated without requiring an unobstructed line of sight, ensuring robustness across diverse occupant postures and driving conditions. Leveraging existing OEM equipment, this approach is both cost-effective and convenient, eliminating the need for additional hardware or specialized installation. Furthermore, with real-time embedded deployment as the ultimate vision, the algorithms were intentionally designed to be minimalistic, prioritizing efficiency over computational complexity. This ensures a fast response time while minimizing the processing power required, making the system well suited for integration into resource-constrained automotive environments where real-time operation is essential. Although the filtered signal amplitude appears low, often in the millivolt range, this is a natural consequence of using seat-integrated sensors not originally designed to capture respiration. Respiration-induced fluctuations are subtle and heavily attenuated by the occupant’s body and seat layers. However, the signal processing framework successfully isolates this weak physiological signal, enabling reliable rate estimation. Importantly, the system’s low signal magnitude does not compromise its practical utility. On the contrary, this solution offers distinct advantages over camera-, radar-, or ultrasound-based methods; it is unaffected by lighting or clothing, requires no line-of-sight or postural alignment, and leverages pre-installed OEM components. These features make it a cost-effective, passive, and robust platform for continuous in-vehicle monitoring, even in dynamic driving conditions.

A key challenge faced in this study was mitigating noise from road-induced vibrations, driver posture shifts, and dynamic vehicle conditions. Despite these confounders, the signal processing framework effectively extracted respiration rates, even in short 20-s windows, where the MAE remained within 2.4 brpm. While this level of precision may not be ideal for high-accuracy clinical monitoring, it remains sufficient for time-sensitive applications such as emergency response scenarios, where rapid estimation is crucial. Shorter windows allow for immediate assessment, while longer windows improve accuracy as more data become available, balancing rapid detection with measurement precision. This adaptability makes the system suitable for both real-time driver monitoring and post-crash triage.

These results demonstrate superior accuracy compared to previous studies that inspired this work. Specifically, Wusk and Gabler [34] explored two methods for respiration rate estimation in a laboratory, with their time-series analysis method yielding a mean error of −2.5 brpm and a standard deviation in error of 2.9 brpm (RPC = 5.69 brpm). Their alternative frequency-based approach reported a slightly lower mean error (−0.91 brpm) but with increased variability (SD = 3.3 brpm, RPC = 6.47 brpm). Valente et al. [18] assessed individual sensors separately in a stationary vehicle with the engine running, reporting mean respiration rates of 18.9 brpm for the reference belt and 21.1 brpm for the pressure-based system, equating to a mean difference of −2.2 brpm. In contrast, the current study achieves a significantly reduced mean difference of −0.05 brpm and standard deviation of 1.87 brpm. Moreover, the reported MAE (1.5 brpm), which was not detailed in previous studies, highlights enhanced precision and reliability. Even the 20-s rapid estimation mentioned earlier reported an MAE of 2.4 brpm, which is comparable to the mean errors of previous studies. It is important to note that the MAE is always greater than or equal to the absolute value of the mean error, suggesting that the mean absolute errors in previous studies would likely have been larger than their reported mean errors. This improved performance, despite the presence of vehicle movement and occupant-induced variability, underscores the robustness and applicability of this system for real-world automotive environments.

Beyond driver monitoring, this technology has broad implications for passenger safety and health applications. Continuous respiration tracking could be particularly useful for infants, children, and elderly passengers, offering a non-intrusive solution for detecting physiological distress. In cases of pediatric vehicular heatstroke, real-time respiration monitoring could alert caregivers before a dangerous situation arises. These findings establish pressure-based respiration sensing as a practical, scalable solution for real-time physiological monitoring in moving vehicles, with applications ranging from everyday driver safety to emergency medical response.

### Limitations and Future Work

While this study demonstrated the feasibility of in-vehicle respiration monitoring, several limitations must be acknowledged. The test vehicle operated at a maximum speed of 35 m.p.h. during data collection. This speed range presents both a constraint and a potential advantage. Higher speeds may introduce additional vehicular vibrations; however, since these are primarily high-frequency disturbances, their impact should be minimal due to the filtering techniques employed. Additionally, a more stable driver posture at higher speeds may reduce body movement artifacts, thereby improving sensor performance. Future research should investigate system performance across a wider range of driving speeds to further assess its robustness in real-world conditions.

Furthermore, this pilot study was limited by its short duration, which may not have captured the full variability of the respiration rates that occur naturally during prolonged driving. Future studies should investigate how respiration variability and sensor accuracy may be affected by longer, fully naturalistic driving sessions. Further research could also explore the effects of secondary activities, such as eating, drinking, speaking, or interacting with dashboard controls, as well as a wider range of road conditions and vehicle types. These scenarios would also introduce a wider range of motion artifacts and environmental disturbances, which are essential for validating the robustness of the filtering algorithm under more complex noise conditions. In addition, larger and more demographically diverse participant samples will be necessary to confirm the generalizability of these findings.

This study validates the system’s reliability in real-world conditions. While the implementation was post hoc, the algorithms were intentionally designed to be minimalistic for real-time operation with limited processing power. The low complexity of the signal processing pipeline supports feasibility for real-time deployment, and the power consumption of the system is minimal due to passive sensing and lightweight operations. More complex methods could improve accuracy but would come at the cost of higher computational demands and slower response times. Future work could explore advanced signal processing techniques such as wavelet transforms, adaptive frequency analysis, empirical mode decomposition, machine-learning-based filtering, or hybrid time–frequency approaches to enhance robustness while maintaining computational efficiency for real-time applications.

While this study focused on technical feasibility, the broader use of in-vehicle physiological monitoring raises important privacy and ethical considerations. Future deployments should ensure that any collected physiological data remain locally processed and are not transmitted or stored externally unless explicitly needed (e.g., to alert emergency services in a crash). Designing such systems with user consent, data minimization, and secure closed-loop architectures can help address potential concerns related to surveillance, misuse, or data sharing.

Beyond validating system robustness, future work should explore additional applications of this technology. While primarily designed for driver monitoring, the system has potential applications in continuous health tracking for passengers, workplace ergonomics, and other domains. Moreover, further analysis of the respiration signal waveforms, including peak characteristics, could provide deeper insights into breathing patterns, depth, and tidal volume variations over time. Deriving additional parameters to quantify respiration quality may offer new avenues for assessing respiratory health and detecting abnormal breathing patterns in various driving conditions.

## 5. Conclusions

This study successfully validates the feasibility of using a pressure-based occupant classification sensor for driver respiration monitoring in a moving vehicle. By addressing challenges such as road-induced vibrations, steering inputs, and body movements, the findings confirm that reliable respiration rate estimation with unobtrusive sensors is possible outside controlled environments. This validation is an important step toward integrating non-intrusive physiological monitoring into vehicles, with implications for driver safety, emergency response, and broader health monitoring applications. Future work can further enhance accuracy and robustness through advanced signal processing techniques, expanding the system’s potential for real-world deployment.

## 6. Patents

The work presented in this study is based upon and contributes to the ongoing development of U.S. Provisional Patent No. 63/697,128, titled “*Seat Sensor-Based Real-Time Respiration Monitoring*”, filed on 20 September 2024. The listed inventors are Sparsh Jain, Jacob Valente, and Miguel Perez. 

## Figures and Tables

**Figure 1 sensors-25-02739-f001:**
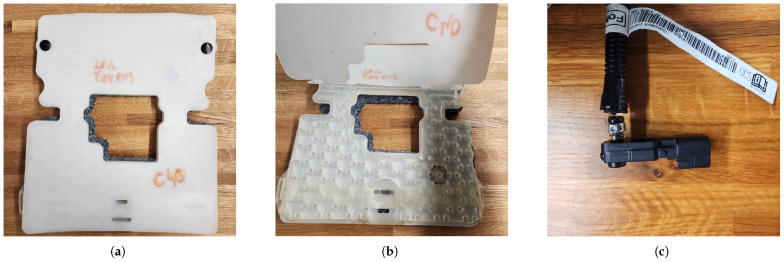
Sensor assembly used: (**a**) sensing mat; (**b**) fluid-filled bladder; (**c**) pressure sensor.

**Figure 2 sensors-25-02739-f002:**
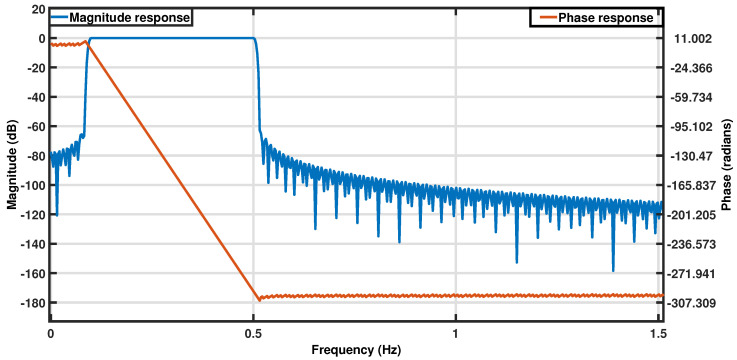
Frequency response of the bandpass filter used.

**Figure 3 sensors-25-02739-f003:**
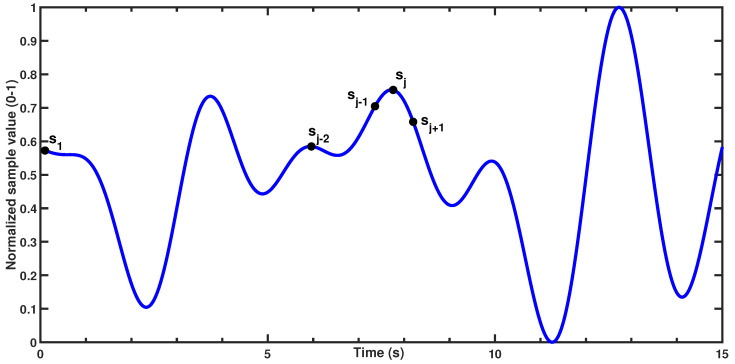
Example of the peak detection algorithm.

**Figure 4 sensors-25-02739-f004:**
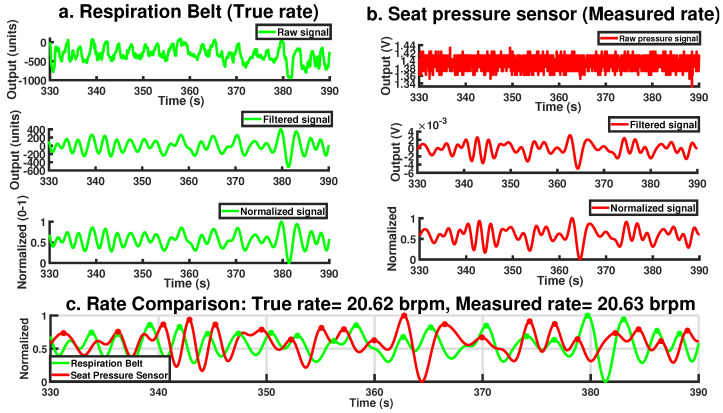
Complete signal processing steps for the (**a**) respiration belt, (**b**) pressure sensor, and (**c**) a comparison of the two normalized signals.

**Figure 5 sensors-25-02739-f005:**
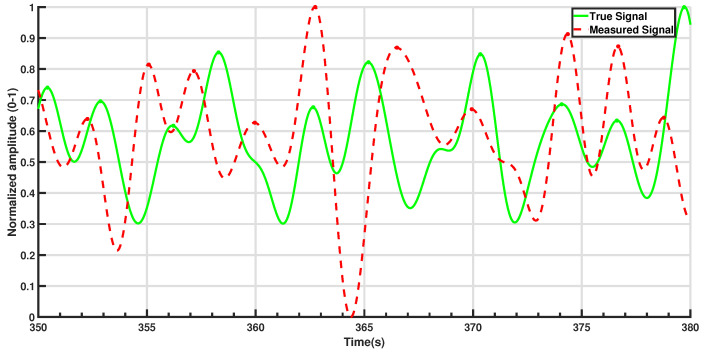
Example of a 30-s epoch for peak comparison. Detected peaks are noted with filled-in circles.

**Figure 6 sensors-25-02739-f006:**
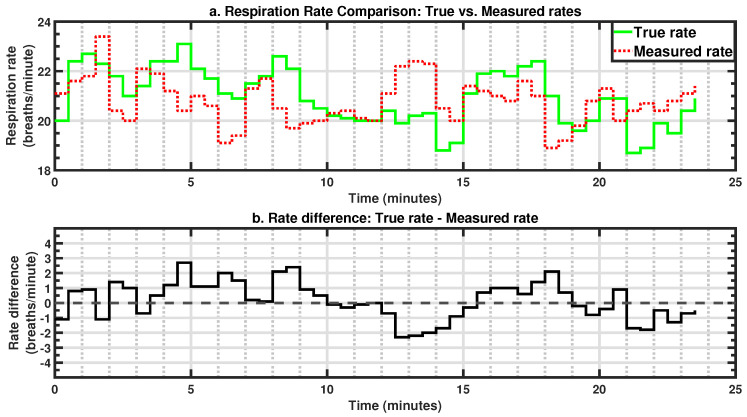
Example: (**a**) Respiration rate comparison over a 25-min period. (**b**) Measurement error.

**Figure 7 sensors-25-02739-f007:**
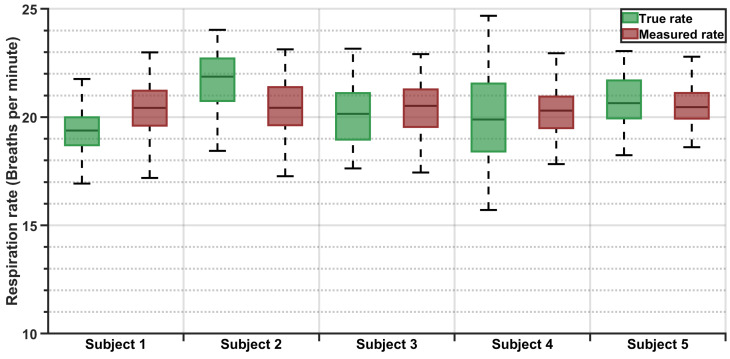
Boxplots for subject-wise respiration rate comparisons.

**Figure 8 sensors-25-02739-f008:**
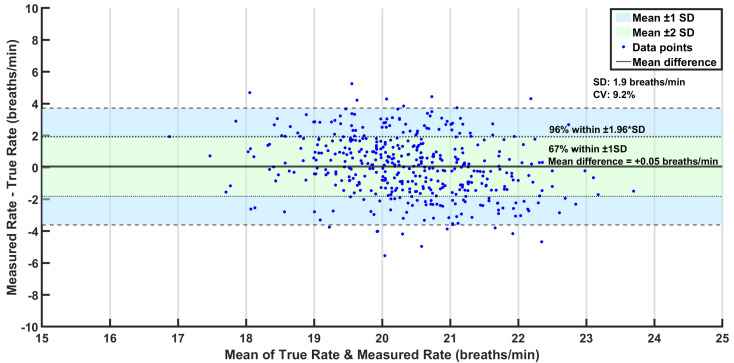
Bland–Altman analysis plot for the dataset.

**Figure 9 sensors-25-02739-f009:**
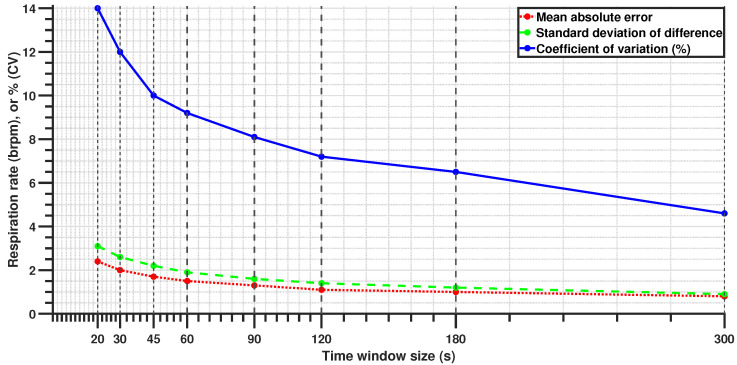
Change in sensor performance with the time-window size (50% overlap).

**Table 1 sensors-25-02739-t001:** OEM specifications for the OCM used in this study.

Parameter	Specification	Notes
Brand/Part	FoMoCo M/N: 12228430	-
Input Supply	5 V DC	-
Output	0–5 V DC	Inverted response
Resolution	2 mV	Estimated

**Table 2 sensors-25-02739-t002:** Filter specifications (designed in MATLAB R2023a).

Parameter	Specification
Response Type	IIR
Method	Elliptic (minimum-order, zero-phase)
Stopband Attenuation	60 dB
Steepness	0.85
Sampling Frequency	50 Hz (Pressure sensor), 300 Hz (Belt)
Passband Frequency	0.1–0.5 Hz (6–30 breaths per minute)

**Table 3 sensors-25-02739-t003:** Window size, overlap, and overlap percentage configurations.

#	Window Size (s)	Overlap (s)	Overlap Percentage (%)
1	20	10	50
2	30	15	50
3	30	20	67
4	45	15	33
5	45	30	67
6	60	15	25
7	60	30	50
8	60	45	75
9	90	45	50
10	90	60	67
11	120	60	50
12	120	90	75
13	180	90	50
14	300	150	50

**Table 4 sensors-25-02739-t004:** Participants’ demographic information.

Subject	Sex	Age (years)	Weight (lbs)	Height (in)
1	Female	35	134.0	65
2	Male	27	172.0	71
3	Male	46	216.0	72
4	Female	36	236.0	62
5	Male	30	163.4	69

**Table 5 sensors-25-02739-t005:** Summary statistics for each subject.

Subject	Minutes	%	Mean	MAE	SD
1	52.5	23.6	−0.9	1.7	1.8
2	41.0	18.4	1.2	1.8	1.9
3	37.0	16.6	−0.3	1.6	1.9
4	41.5	18.6	−0.2	1.5	1.9
5	50.5	22.7	0.2	1.1	1.3

**Table 6 sensors-25-02739-t006:** Performance metrics across different window sizes and overlaps.

#	Window Size (s)	Overlap (s)	Overlap %	Mean (brpm) *	MAE (brpm)	SD (brpm)	CV (%)
1	20	10	50	−0.02	2.4	3.1	14.0
2	30	15	50	−0.04	2.0	2.6	12.0
3	30	20	67	−0.03	2.0	2.6	12.0
4	45	15	33	−0.04	1.7	2.2	10.0
5	45	30	67	−0.07	1.7	2.2	10.0
6	60	15	25	−0.05	1.5	1.9	9.4
7	60	30	50	−0.05	1.5	1.9	9.2
8	60	45	75	−0.02	1.5	1.9	9.5
9	90	45	50	0.03	1.3	1.6	8.1
10	90	60	67	0.08	1.2	1.6	7.9
11	120	60	50	0.07	1.1	1.4	7.2
12	120	90	75	0.08	1.1	1.4	7.1
13	180	90	50	0.10	1.0	1.2	6.5
14	300	150	50	−0.10	0.8	0.9	4.6

* All values are computed for True rate-Measured rate.

## Data Availability

The raw data supporting the conclusions of this article will be made available by the authors upon request.

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
