# Peer review of "On-Road Evaluation of an Unobtrusive In-Vehicle Pressure-Based Driver Respiration Monitoring System"

_sensors, 2025, doi:10.3390/s25092739_

Round 1

Reviewer 1 Report

Comments and Suggestions for Authors

This article presents a compelling pilot study that evaluates the feasibility and accuracy of a pressure-based respiration monitoring system embedded in a vehicle seat. The main objective is to determine whether such a non-intrusive setup can reliably estimate driver respiration rate during real-world driving conditions, thus contributing to applications such as post-crash triage, impairment detection, and continuous health monitoring.

The study addresses a relevant question, especially in the context of advancing vehicle automation and driver monitoring systems. The integration of a cost-effective, OEM-based pressure sensor system into the car seat is a particularly innovative aspect, as it eliminates the need for additional intrusive hardware while leveraging existing vehicle components. This contributes significantly to the field, especially considering the growing interest in unobtrusive in-cabin sensing technologies.

The experimental design is clearly described and methodologically sound. The authors implement a rigorous signal processing pipeline and compare the pressure sensor’s output with a reference respiration belt under dynamic driving conditions. The statistical analysis is thorough, employing mean absolute error (MAE), standard deviation, and Bland-Altman plots to assess agreement between the two measurements. Results show promising accuracy, with a mean absolute error of 1.5 brpm and a standard deviation of 1.87 brpm. These findings are impressive given the complexity of the driving environment.

One of the strongest aspects of the manuscript is the careful comparison with previous studies, which highlights the advancement made by this work in terms of both accuracy and real-world applicability. The use of different time windows for estimation and their impact on accuracy adds depth to the analysis and practical insight for real-world deployment scenarios.

The conclusions are well supported by the results and directly respond to the research question. The authors convincingly argue for the feasibility of integrating such systems in commercial vehicles. Moreover, they appropriately identify the limitations of the current study—such as limited driving speed, short session durations, and a small participant sample—while offering concrete suggestions for future research, including testing across diverse driving contexts, extended use cases (e.g., pediatric safety), and the integration of advanced signal processing or machine learning techniques.

However, there are a few areas for improvement. Although the sample size is sufficient for a pilot study, further studies with a more diverse and larger sample would be necessary to generalize the results. Lastly, it would be valuable to see a brief discussion of potential privacy and ethical concerns related to physiological monitoring in vehicles.

The references are appropriate and up-to-date, encompassing foundational studies in the field as well as recent technological advancements.

Author Response

Reviewer #1:

This article presents a compelling pilot study that evaluates the feasibility and accuracy of a pressure-based respiration monitoring system embedded in a vehicle seat. The main objective is to determine whether such a non-intrusive setup can reliably estimate driver respiration rate during real-world driving conditions, thus contributing to applications such as post-crash triage, impairment detection, and continuous health monitoring.

The study addresses a relevant question, especially in the context of advancing vehicle automation and driver monitoring systems. The integration of a cost-effective, OEM-based pressure sensor system into the car seat is a particularly innovative aspect, as it eliminates the need for additional intrusive hardware while leveraging existing vehicle components. This contributes significantly to the field, especially considering the growing interest in unobtrusive in-cabin sensing technologies.

The experimental design is clearly described and methodologically sound. The authors implement a rigorous signal processing pipeline and compare the pressure sensor’s output with a reference respiration belt under dynamic driving conditions. The statistical analysis is thorough, employing mean absolute error (MAE), standard deviation, and Bland-Altman plots to assess agreement between the two measurements. Results show promising accuracy, with a mean absolute error of 1.5 brpm and a standard deviation of 1.87 brpm. These findings are impressive given the complexity of the driving environment.

One of the strongest aspects of the manuscript is the careful comparison with previous studies, which highlights the advancement made by this work in terms of both accuracy and real-world applicability. The use of different time windows for estimation and their impact on accuracy adds depth to the analysis and practical insight for real-world deployment scenarios.

The conclusions are well supported by the results and directly respond to the research question. The authors convincingly argue for the feasibility of integrating such systems in commercial vehicles. Moreover, they appropriately identify the limitations of the current study—such as limited driving speed, short session durations, and a small participant sample—while offering concrete suggestions for future research, including testing across diverse driving contexts, extended use cases (e.g., pediatric safety), and the integration of advanced signal processing or machine learning techniques.

However, there are a few areas for improvement.

We thank the reviewer for their thoughtful and constructive feedback. We carefully considered each comment and revised the manuscript accordingly. Below we provide a detailed response to each point, indicating how and where changes were made.

Comment 1: Although the sample size is sufficient for a pilot study, further studies with a more diverse and larger sample would be necessary to generalize the results.

Response: We appreciate this comment. As noted by the reviewer, the need for a larger and more demographically diverse sample is now explicitly acknowledged in the revised manuscript. Specifically, we have added a sentence in the Limitations and Future Directions paragraph (Section 4, pg. 13, line 338) emphasizing that future studies should include more participants across a broader demographic spectrum to improve the generalizability of the findings.

Comment 2: Lastly, it would be valuable to see a brief discussion of potential privacy and ethical concerns related to physiological monitoring in vehicles.

Response: Thank you for this important suggestion. We have added a paragraph under the Limitations and Future Directions paragraph (Section 4 pg. 13, line 352) that briefly discusses privacy and ethical considerations related to in-vehicle physiological monitoring. We note that while some systems could raise concerns regarding data collection and transmission, our proposed implementation could be deployed in a closed-loop, local-only processing format. This would ensure that physiological data remains within the vehicle system and is not externally transmitted, except in emergency scenarios such as EMS notifications.

We thank the reviewer again for their careful reading and helpful insights, which have strengthened the manuscript. We hope the revised version meets the expectations and standards of the journal.

Reviewer 2 Report

Comments and Suggestions for Authors

This manuscript proposed a driver breathing monitoring system based on a pressure sensor. The experiments are conducted in a real driving environment, taking into full account practical challenges such as dynamic noise. The experimental results support the main conclusions of the paper. However, there is still room for further improvement in some aspects.

  1. Although the filtering algorithm used in the manuscript performs well under experimental conditions, its adaptability to more complex noise has not been verified.
  2. In this study, how is the sampling frequency of 50 Hz optimized for breath signal acquisition?
  3. Although the proposed system is expected to achieve continuous monitoring, the manuscript does not discuss real-time processing feasibility, computational complexity, or power consumption considerations.
  4. The manuscrip lacks parameters such as driver age, weight, and health status, which may affect the results.
  5. The raw data output voltage in Figure 4b varies within 10-1V, and the filtered signal voltage varies within 10-3V. The signal-to-noise ratio seems to be very low. Please explain the advantages of this solution compared to other respiratory monitoring solutions.

Author Response

Reviewer #2:

This manuscript proposed a driver breathing monitoring system based on a pressure sensor. The experiments are conducted in a real driving environment, taking into full account practical challenges such as dynamic noise. The experimental results support the main conclusions of the paper. However, there is still room for further improvement in some aspects.

We thank the reviewer for their thoughtful and constructive feedback. We carefully considered each comment and revised the manuscript accordingly. Below we provide a detailed response to each point, indicating how and where changes were made.

Comment 1: Although the filtering algorithm used in the manuscript performs well under experimental conditions, its adaptability to more complex noise has not been verified.

Response: We appreciate this comment and fully agree with the reviewer’s observation. Section 4 (pg. 13, line 336) of the Limitations and Future Work section explicitly acknowledges this limitation. We note that future research should investigate the robustness of the system across longer, more naturalistic driving sessions that include diverse sources of noise, such as occupant motion, secondary tasks, and varying road conditions. These efforts will be essential to validate the system’s generalizability and performance under more complex and realistic environments.

Comment 2: In this study, how is the sampling frequency of 50 Hz optimized for breath signal acquisition?

Response: Thank you for raising this point. A clarification has been added in Section 2.2 (pg. 4, line 126) to explain that the 50 Hz sampling rate was selected to comfortably exceed the Nyquist rate for respiratory signals, which fall within 0.1-0.5 Hz for normal breathing. The high sampling rate also served secondary purposes, including synchronization with other vehicle systems and capturing higher-frequency disturbances for potential future analyses.

Comment 3: Although the proposed system is expected to achieve continuous monitoring, the manuscript does not discuss real-time processing feasibility, computational complexity, or power consumption considerations.

Response: We agree this is an important aspect. We have added 1 sentence in Section 4 (pg. 13, line 343) stating that our signal processing pipeline was intentionally kept simple to enable low-latency, real-time feasibility in future implementations. We also note that the computational and power requirements are minimal due to the lightweight filtering operations and low data volume.

Comment 4: The manuscript lacks parameters such as driver age, weight, and health status, which may affect the results.

Response: We appreciate this comment and have addressed it in the revised manuscript. Specifically, we have added a description of the subjects and new demographic table (Table 4, Section 3, pg. 7-8, Line 206) summarizing the age, sex, weight, and height of all five participants.

All participants were healthy adults. The participant weights span a range from approximately the 20th–9st percentile for females and the 23rd–71st percentile for males, providing reasonable coverage of the expected adult population. While individual factors like age or weight may influence baseline respiration rates, these variations remain within the target detection band of 6–30 breaths per minute. The system is designed to function reliably across this range, and performance is not expected to be directly affected by demographic parameters unless the sensor lacks sufficient sensitivity or adaptability, which was not observed. We also refer the reviewer to our previous publication [https://doi.org/10.1016/j.aap.2023.107302], where sensor performance was evaluated across a broader anthropometric range under static conditions.

Comment 5: The raw data output voltage in Figure 4b varies within 10^-1 V, and the filtered signal within 10^-3 V. The signal-to-noise ratio seems to be very low. Please explain the advantages of this solution compared to other respiratory monitoring solutions.

Response: We appreciate this observation. A clarification has been added in Section 2.2 (pg. 5, line 153) to explain that while the filtered signal appears low in amplitude, this is expected and not indicative of low-quality sensing. The respiration-induced variations are inherently subtle and heavily damped by the body and seat system. Moreover, SNR in this context is reversed compared to typical sensing scenarios- the tiny oscillations are typically treated as noise in the original use case (occupant classification) but are signal here. We have also explicitly reiterated the system’s advantages in the Discussion (Section 4, pg. 12, line 274): passive sensing, robust under lighting and environmental variability, and cost-effective integration into existing vehicle systems.

We thank the reviewer again for their careful reading and helpful insights, which have strengthened the manuscript. We hope the revised version meets the expectations and standards of the journal.

Reviewer 3 Report

Comments and Suggestions for Authors

The paper is about driver monitoring and automotive safety detection. The authors use fluid-filled pressure bladder sensor to solve the detection problem. The paper is interesting but needs minor improvements.

(1) some keywords may not be closely related to the paper, e.g., pediatric safety.

(2) in Section 4, subsection 4.1 is not necessary because section 4.2 does not exist.

(3) In figure 5, the curves are not clear enough compared with other figures. It is better to improve the quality of the figure or increase the size of the line.

(4) There are many configurations in table 3. After experiment, which is the best one for recommendation in the system.

Author Response

Reviewer #3

The paper is about driver monitoring and automotive safety detection. The authors use fluid-filled pressure bladder sensor to solve the detection problem. The paper is interesting but needs minor improvements.

We thank the reviewer for their thoughtful and constructive feedback. We carefully considered each comment and revised the manuscript accordingly. Below we provide a detailed response to each point, indicating how and where changes were made.

Comment 1: Some keywords may not be closely related to the paper, e.g., pediatric safety.

Response: We agree. The keyword "pediatric safety" has been removed and replaced with “in-vehicle sensing” (see Pg. 1)

Comment 2: In Section 4, subsection 4.1 is not necessary because section 4.2 does not exist.

Response: Thank you for this observation. Subsection 4.1 has been removed, and the section has been retitled accordingly (Section 4, pg. 13, line 321).

Comment 3: In Figure 5, the curves are not clear enough compared with other figures. It is better to improve the quality of the figure or increase the size of the line.

Response: We have increased the line thickness in Figure 5 to improve visual clarity while retaining the original high resolution (Figure 5, pg. 8).

Comment 4: There are many configurations in Table 3. After experiment, which is the best one for recommendation in the system?

Response: We appreciate the question. We have clarified in Section 2.3.1. (pg. 7, line 196) that the 60-second window with 50% overlap was selected as a practical compromise for steady-state monitoring. We also reiterate in Section 4 how shorter windows may suit emergency scenarios, while longer windows provide smoother tracking for continuous monitoring. Table 3 configurations were explored to evaluate sensitivity to parameter choice rather than to determine a single best configuration.

We thank the reviewer again for their careful reading and helpful insights, which have strengthened the manuscript. We hope the revised version meets the expectations and standards of the journal.

Round 2

Reviewer 2 Report

Comments and Suggestions for Authors

The revisions meet my requirments.